# Expression signature based on TP53 target genes doesn't predict response to TP53-MDM2 inhibitor in wild type TP53 tumors

Dmitriy Sonkin*

Division of Cancer Treatment and Diagnosis, National Cancer Institute, Rockville, United States

**Abstract** A number of TP53-MDM2 inhibitors are currently under investigation as therapeutic agents in a variety of clinical trials in patients with TP53 wild type tumors. Not all wild type TP53 tumors are sensitive to such inhibitors. In an attempt to improve selection of patients with TP53 wild type tumors, an mRNA expression signature based on 13 TP53 transcriptional target genes was recently developed (Jeay et al. 2015). Careful reanalysis of TP53 status in the study validation data set of cancer cell lines considered to be TP53 wild type detected TP53 inactivating alterations in 23% of cell lines. The subsequent reanalysis of the remaining TP53 wild type cell lines clearly demonstrated that unfortunately the 13-gene signature cannot predict response to TP53-MDM2 inhibitor in TP53 wild type tumors.

*For correspondence: dmitriy.sonkin@nih.gov

**Competing interests:** The author declares that no competing interests exist.

## Introduction

A number of TP53-MDM2 inhibitors are currently under investigation as therapeutic agents in a variety of clinical trials across multiple tumor types. Mechanistically, only tumors with wild-type (WT) TP53 can potentially be sensitive to TP53-MDM2 inhibitors as confirmed in part by sensitivity of WT MEFs cells and by the loss of sensitivity in TP53 knockout MEFs (*Efeyan et al., 2007*). Therefore, clinical trials of TP53-MDM2 inhibitors only include patients with WT TP53 tumors. Based on pre-clinical work, it is clear that not all WT TP53 tumors are sensitive to TP53-MDM2 inhibitors. Multiple attempts have been made to try to predict sensitivity to TP53-MDM2 inhibitors in WT TP53 tumors. Unfortunately, despite these efforts, there is currently no clinically validated and FDA-approved assay to identify WT TP53 tumors most likely to respond to TP53-MDM2 inhibitors.

Recently, *Jeay et al., (2015)* attempted to find a messenger RNA (mRNA) predictive expression signature to selective TP53-MDM2 inhibitor NVP-CGM097 using a panel of cell lines from the Cancer Cell Line Encyclopedia (CCLE) (*Barretina et al., 2012*) with corresponding genetic and genomic datasets. As a result of this work, *Jeay et al., (2015)* described the mRNA signature based on 13 TP53 transcriptional target genes. The signature was generated using TP53-MDM2 inhibitor sensitive versus insensitive cell lines without regard to the TP53 status. As a critical part of the validation work, *Jeay et al., (2015)* used an independent set of 52 cancer cell lines that were considered to be TP53 WT. Since the signature was generated without considering TP53 status and the fact that TP53-MDM2 inhibitors can only be effective in WT TP53 tumors, the signature is likely to represent a proxy for TP53 status. Therefore, it would not be expected to enhance the ability to predict sensitivity to TP53-MDM2 inhibitors in TP53 WT tumors, so the reported predictive ability of the *Jeay et al., (2015)* signature in the set of 52 cancer cell lines considered by authors as TP53 WT is surprising.

**eLife digest** Damaged cells in the human body can develop into tumors if left unchecked. TP53 (also called p53) is a protein that normally helps to repair or eliminate these damaged cells and prevent tumors from forming. About half of all cancerous tumors have mutations that prevent TP53 from working. In tumors with normal TP53 (called TP53 wild type tumors), another protein that acts to keep TP53 in check is often overly active. This overactive protein (called MDM2) prevents TP53 from suppressing tumor development. Many scientists are developing anticancer drugs called TP53-MDM2 inhibitors to target the potentially overactive protein in TP53 wild type tumors, and importantly only a tumor with working TP53 would have a chance of responding to this kind of inhibitor.

Earlier in 2015, a team of researchers at the Novartis Institutes for BioMedical Research reported the results of a screen of hundreds of cancer cell lines that investigated which ones were sensitive to TP53-MDM2 inhibitors. Using mix of TP53 mutant and TP53 wild type cancer cell lines, the Novartis team identified a set of 13 genes that were highly expressed in cell lines that were sensitive to one of these inhibitors. This 13-gene signature was then suggested as a way to identify which cancer patients with TP53 wild type tumors would be the most likely to benefit from treatment with TP53-MDM2 inhibitors.

However, now Dmitriy Sonkin has reanalyzed the validation set of TP53 wild type cancer cell lines used by the Norvartis team and found that many of them had been mistakenly identified as TP53 wild type. That is to say around a quarter of the cell lines thought to have normal TP53 actually had mutations in the gene for TP53. Sonkin then repeated the analysis using only those cell lines that were from TP53 wild type tumors. This revealed that the 13-gene signature cannot predict how cancer cells from a TP53 wild type tumor will respond to a TP53-MDM2 inhibitor. Further work would be beneficial in order to find an accurate test to determine which cancer patients will benefit the most from treatment with TP53-MDM2 inhibitors.

## Results and discussion

One potential explanation for the reported predictive ability of the *Jeay et al., (2015)* signature in a validation set of 52 cancer cell lines that were considered TP53 WT is the possibility that some of these cell lines have TP53 inactivating alterations that were missed during cell lines selection. TP53 could be inactivated by a variety of mechanisms including inactivating mutations, DNA loss and loss of mRNA expression. The CCLE provides sequencing, copy number and mRNA expression data, enabling careful examination of TP53 status in the set of 52 cancer cell lines used for validation by *Jeay et al., (2015)*. Careful examination of TP53 status using publicly available CCLE mutation calls, copy number and mRNA expression (described in Materials and methods) identified 12 out of 52 cancer cell lines containing inactivating TP53 alterations, which are summarized in *Table 1*.

As can be seen in *Table 1*, the majority of 12 cell lines have inactivating TP53 point mutations, three cell lines with TP53 frame shift mutations exhibit loss of TP53 mRNA expression likely due to nonsense-mediated mRNA decay, two other cell lines also have loss of TP53 mRNA expression. (Gene expression and Copy Number (CN) cutoffs are defined in Materials and methods). Importantly, since only TP53 WT tumors have a chance of being sensitive to TP53-MDM2 inhibitors, all 12 cell lines are insensitive to NVP-CGM097.

In order to re-evaluate the performance of the signature in TP53 WT settings, the 12 cancer cell lines with inactivated TP53 listed in *Table 1* have been removed from the *Jeay et al., (2015)* validation list of cell lines, resulting in set of 40 likely WT cancer cell lines listed in *Supplementary file 1A* with information on sensitivity to NVP-CGM097 and *Jeay et al., (2015)* 13-gene signature prediction. Results of reevaluation of signature performance are listed in *Table 2*.

As can be seen from *Table 2*, the 13-gene signature positive predicted value for NVP-CGM097 does not noticeably differ from the response rate to the inhibitor. Also the specificity and negative predicted value (NPV) are low and it is likely that the actual specificity and NPV are even lower considering that DAN-G cell line has TP53 mRNA expression just above cutoff for TP53 mRNA loss. (NVP-CFC218 is another TP53-MDM2 inhibitor used by *Jeay et al., (2015)* that is structurally and

**Table 1.** List of 12 cell lines with inactivated TP53 in the validation set of 52 cancer cell lines considered to be TP53 wild-type by *Jeay et al., (2015)*.

| Cell line name | TP53 inactivating mutation(s) | Alternative reads/reference reads | TP53 mRNA (MAS5-150 201746_at) | TP53 CN ratio | Jeay et al. 13-gene signature prediction | NVP-CGM097 sensitivity |
|---|---|---|---|---|---|---|
| KASUMI-1 | p. R248Q | 52/0 | 265 | 0.54 | insensitive | insensitive |
| COLO-818 | p. C135R | 34/0 | 257 | 1.14 | insensitive | insensitive |
| IGR-37 | p. C229fs | 110/11 | 9 | 0.59 | insensitive | insensitive |
| HCC202 | p. T284fs | 35/4 | 14 | 0.8 | insensitive | insensitive |
| EFM-192A | p. F270fs | 7/1 | 10 | 0.74 | insensitive | insensitive |
| NCI-H1568 | p. H179R | 89/1 | 202 | 0.82 | insensitive | insensitive |
| COLO-783 | p. P27L | 38/0 | 304 | 1.05 | sensitive | insensitive |
| GA-10 | p. I232N, p. P152L | 94/50, 52/76 | 493 | 0.81 | insensitive | insensitive |
| VMRC-RCW | p. I332_splice | 192/68 | 63 | 1.65 | insensitive | insensitive |
| JHH-5 | p. PPQH190del | 107/41 | 272 | 1.03 | insensitive | insensitive |
| HDLM-2 | | | 1 | 0.94 | insensitive | insensitive |
| RERF-LC-KJ | | | 25 | 1.3 | insensitive | insensitive |

biochemically very similar to NVP-CGM097 and, as can be seen from *Supplementary file 1B*, signature has the same pattern of performance for NVP-CFC218 as for NVP-CGM097).

*Figure 1* provides a visual overview of data used for evaluating the 13-gene signature performance for NVP-CGM097 in the validation set of 40 likely TP53 WT cancer cell lines. *Figure 1* clearly illustrates that the *Jeay et al., (2015)* 13-gene signature cannot differentiate between sensitive and insensitive cell lines. Keeping in mind that DAN-G cell line may have TP53 mRNA loss, there is only one cell line, HCC-95, that is correctly predicted to be insensitive. The presented reanalysis of signature performance in this article strongly suggests that 13-gene signature is a proxy for TP53 status. In such case, one can put forward the hypothesis that HCC-95 may harbor an inactivating alteration (s) that has been missed. If this is the case, the *Jeay et al., (2015)* 13-gene signature has zero specificity and zero NPV in the validation set of TP53 WT cancer cell lines.

**Table 2.** Performance of *Jeay et al., (2015)* 13-gene signature prediction in validation set of 40 likely TP53 wild-type cancer cell lines.

| Performance measure | Cell sensitivity defined by NVP-CGM097 |
|---|---|
| Sensitivity | 89% (24/27) |
| Specificity | 15% (2/13) {DAN-G removal 8% (1/12) *} |
| PPV | **68.6%** (24/35) |
| NPV | 40% (2/5) {DAN-G removal 25% (1/4) *} |
| Response rate | **67.5%** (27/40) |

* DAN-G has TP53 mRNA expression of 33 (MAS5-150 201746_at) indicating the probable loss of TP53 mRNA. (Stringent TP53 mRNA expression cutoff is set at 32 (MAS5-150 201746_at) to indicate loss of TP53 mRNA). NPV - negative predicted value; PPV - positive predicted value.

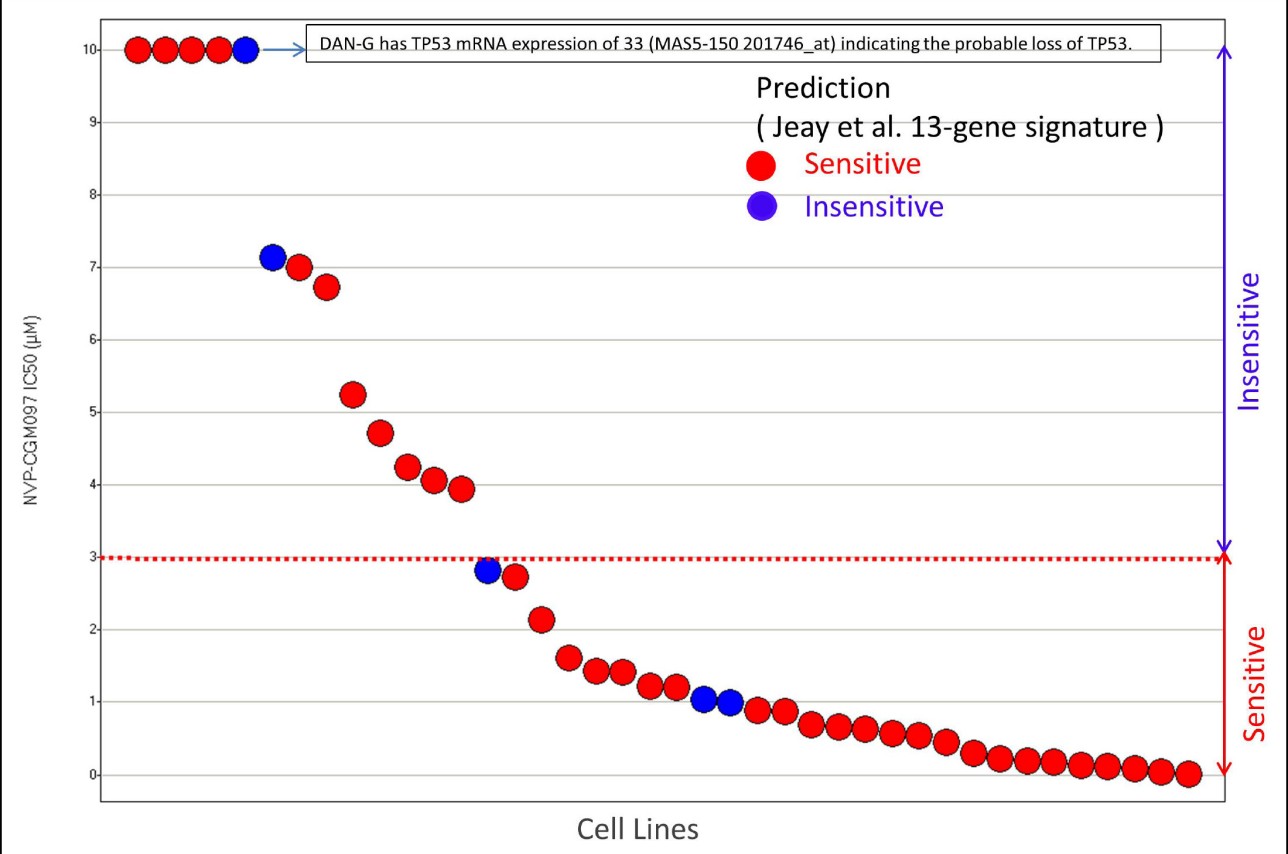

**Figure 1.** Cell lines sensitivity to NVP-CGM097 in validation set of 40 likely TP53 WT cell lines.

In clinical sequencing, special care is taken to make sure sufficient coverage is obtained across all target regions in order to reliably detect point mutations, insertions/deletions, fusions and copy number aberrations (*Frampton et al., 2013*). Often the additional step of manual review of sequencing analysis results is added to the workflow to detect false positive/negative calls due to particular sequence composition or computational pipeline artifacts. In preclinical sequencing, such detailed workflow is often too expensive to obtain. In the CCLE, RainDance technology (*Mazutis et al., 2009*) was used to fill some blind spots in the hybrid capture process, but such a process does not necessarily mitigate all problematic regions. This may explain the potentially missed TP53 inactivating alteration(s) in HCC-95.

In addition to providing sensitivity to NVP-CGM097 in the validation set, *Jeay et al., (2015)* also provided sensitivity data to NVP-CFC218 in the set of 356 CCLE cancer cell lines. In this set, *Jeay et al., (2015)* reported the presence of 113 cell lines with WT TP53 with response rate of 38% to NVP-CFC218. Based on the detection of 12 cell lines with inactivated TP53 in the validation set of 52 cell lines considered to be TP53 WT by *Jeay et al., (2015)*, a careful examination of TP53 status in the set of 113 cell lines considered to be TP53 WT resulted in identification of 29 cancer cell lines containing inactivating TP53 alterations, which are summarized in *Supplementary file 1C*. Importantly, since only TP53 WT tumors have a chance of being sensitive to TP53-MDM2 inhibitors, all 29 cell lines are insensitive to NVP-CFC218. Sensitivity to NVP-CFC218 in the remaining 84 likely TP53 WT cell lines is summarized in *Supplementary file 1D*. Analysis of this data indicates the response rate of 51% (43/84).

Based on the current analysis, it is clear that the *Jeay et al., (2015)* 13-gene signature is a proxy for TP53 status. It has a good, but of course not perfect, ability to detect cell lines with inactivated TP53 and this ability could be useful in some of the preclinical work. For example, it could be useful to look at the cell lines that are insensitive to TP53-MDM2 inhibitor(s) and also predicted by

*Jeay et al., (2015)* 13-gene signature to be insensitive, but not annotated as TP53 inactivated; it is likely that significant fraction of such cell lines harbor undetected TP53 inactivating alterations.

In summary, it is clear that *Jeay et al., (2015)* 13-gene signature unfortunately cannot predict response to TP53-MDM2 inhibitor in TP53 WT tumors. Therefore the ability to predict sensitivity to TP53-MDM2 inhibitors in WT TP53 tumors is still out of reach. The development of such prediction capacity would be clinically beneficial and also may provide valuable insights into the understanding of some of important areas of cancer biology.

## Materials and methods

NVP-CGM097 and NVP-CFC218 pharmacologic cell line profiling data, 13-gene signature predictions have been obtained from (*Jeay et al., 2015*).

Affymetrix U133Plus2 mRNA expression, Affymetrix SNP 6.0 data, OncoMap mutation calls (*MacConaill et al., 2009*), exome sequencing data (*Hodges et al., 2007*) have been obtained from CCLE website (http://www.broadinstitute.org/ccle/home). TP53 mutation calls have been obtained from the following two files: CCLE_hybrid_capture1650_hg19_NoCommonSNPs_NoNeutralVariants_CDS_2012.05.07.maf (22-May-2012) and 1650_HC_plus_RD_muts.maf.annotated (24-Nov-2014), both files are available for download from CCLE website. Genomic characterization section in Supplementary methods (*Barretina et al., 2012*) provides a detailed description of sequencing data generation and variant calling pipeline. *Supplementary file 1E* provides COSMIC information on TP53 mutations in cell lines listed in the *Table 1* and also includes COSMIC sample ID for each of the cell lines.

Copy number (CN) ratio is the ratio of signal intensity in a tumor sample versus normal reference samples normalized to total DNA quantity; thus a CN ratio of 1 corresponds to a diploid locus. CN ratio <0.6 indicates 'allelic loss'. CN ratio <0.25 indicates 'bi-allelic loss'.

All mRNA expression values are MAS5 normalized, with a 2% trimmed mean of 150 (*Hubbell, Liu, and Mei 2002*). TP53 Affymetrix (201746_at) mRNA MAS5-150 normalized expression values below 32 are considered to be indicative of TP53 'mRNA loss'.

## Additional information

### Funding

The author declares that there was no funding for this work.

### Author contributions

DS, Conception and design, Analysis and interpretation of data, Drafting or revising the article

## Additional files

### Supplementary files

• Supplementary file 1. (A) 40 likely TP53 WT cancer cell lines from Jeay et al. (2015) validation set. (BPerformance of Jeay et al. (2015) 13-gene Signature Prediction in validation set of 28 likely TP53 WT cancer cell lines with sensitivity defined by NVP-CFC218. (NVP-CFC218 data was available for 28 out of 40 likely TP53 WT cancer cell lines.) (CList of 29 cell lines with inactivated TP53 in the set of 113 cancer cell lines considered to be TP53 wild type by Jeay et al. (2015). (D) 84 likely TP53 WT cancer cell lines from Jeay et al. (2015). (E) COSMIC information on mutations in cell lines listed in the Table 1.

### Major datasets

The following previously published dataset was used:

| Author(s) | Year | Dataset title | Dataset URL | Database, license, and accessibility information |
|---|---|---|---|---|
| Barretina J, Caponigro G, Stransky N, Venkatesan K | 2012 | SNP and Expression data from the Cancer Cell Line Encyclopedia (CCLE) | http://www.ncbi.nlm.nih.gov/geo/query/acc.cgi?acc=GSE36139 | Publicly available at the NCBI Gene Expression Omnibus (Accession no: GSE36139). |

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
