## [Decision Letter]

Thank you for submitting your work entitled "Expression signature based on TP53 target genes doesn't predict response to TP53-MDM2 inhibitor in wild type TP53 tumors" for consideration by *eLife*. Your article has been reviewed by two peer reviewers, and the evaluation has been overseen by a Reviewing Editor and Tony Hunter as the Senior Editor.

The reviewers have discussed the reviews with one another and the Reviewing editor has drafted this decision to help you prepare a revised submission.

The current report by Sonkin challenges the conclusions of a recent paper by Jeay et al. reporting the discovery of a gene signature that could potentially predict the efficacy of small molecule inhibitors of the p53-MDM2 interaction in the treatment of tumors expressing wild-type p53. Upon reanalysis of the p53 mutational status of the cell lines employed, Sonkin determined that Jeay et al. had not assessed the mutational status of p53 correctly for a number of cell lines. When the analysis is repeated with the annotation by Sonkin, the gene signature no longer predicts drug efficacy, simply discriminating between wild-type and mutant cell lines. In this regard, the report by Sonkin is important, as it challenges some of the main conclusions in the report by Jeay et al. about the utility of the gene signature.

Having said that, it is surprising and unfortunate that neither the original nor the present paper has a clear description of how the variants were called. The original paper says, in the Materials and methods that "gene level genetic alterations […] were compiled from the Sanger center COSMIC data and internal sources including Exome Capture Sequencing of 1600 cancer related genes", but it does not say exactly what methods were used to call and/or validate variants. Likewise, the present paper simply states: "Careful examination of TP53 status identified 12 out of 52 cancer cell lines containing TP53 alterations…".

It is then necessary for Sonkin to provide additional evidence for the mutations he found, just so that the reader can better understand why Jeay et al. missed them. Based on Table 1, these mutations have a lot of reads with alternative alleles, meaning that most algorithms should have caught them. Why did Jeay et al. miss them? Are they in hard-to-capture regions? Perhaps IGV screenshots could be provided in the supplement (along with GC content or read density map from some exomes to show hard-to-capture areas?). What exactly is the 'careful examination' performed by Sonkin? Also, is Sonkin proposing that the information in the Sanger Center and COSMIC databases about p53 mutational status of these cell lines is wrong? Is the method by Sonkin challenging the annotation in these databases and should these repositories be notified about these potential mistakes?

In brief, a more thorough discussion of the various methodologies employed for variant calling and an effort to explain why Jeay et al. and the public databases are producing different results from those obtained by Sonkin would be required to advance publication of this work, whose potential impact is very high, as it would not only challenge the paper by Jeay et al., but also public databases employed by thousands of researchers around the world.

---

## [Author Response]

*[…] Having said that, it is surprising and unfortunate that neither the original nor the present paper has a clear description of how the variants were called. The original paper says, in the Materials and methods that "gene level genetic alterations […] were compiled from the Sanger center COSMIC data and internal sources including Exome Capture Sequencing of 1600 cancer related genes", but it does not say exactly what methods were used to call and/or validate variants. Likewise, the present paper simply states: "Careful examination of TP53 status identified 12 out of 52 cancer cell lines containing TP53 alterations…".*

All cell lines referenced in the manuscript were part of Cancer Cell Line Encyclopedia (CCLE) and therefore I used publicly available CCLE mutation calls (Barretina et al., 2012). The manuscript was modified to make it clear that publicly available TP53 CCLE mutation calls have been used (Results and Discussion). I’ve modified the Materials and methods to reference exact two CCLE files from CCLE website which were used as source of TP53 mutation calls and also provided reference to genomic characterization section provided by Barretina et al., which includes detailed description of sequencing data generation and variant calling pipeline.

It is then necessary for Sonkin to provide additional evidence for the mutations he found, just so that the reader can better understand why Jeay et al. missed them. Based on Table 1, these mutations have a lot of reads with alternative alleles, meaning that most algorithms should have caught them. Why did Jeay et al. miss them? Are they in hard-to-capture regions? Perhaps IGV screenshots could be provided in the supplement (along with GC content or read density map from some exomes to show hard-to-capture areas?). What exactly is the 'careful examination' performed by Sonkin? Also, is Sonkin proposing that the information in the Sanger Center and COSMIC databases about p53 mutational status of these cell lines is wrong? Is the method by Sonkin challenging the annotation in these databases and should these repositories be notified about these potential mistakes?

Indeed Sanger Institute COSMIC databases are extremely valuable resource and used by thousands of researchers around the world. COSMIC database contains sequencing data for 8 out of 10 cell lines with TP53 mutations listed in Table 1 of my manuscript containing cell lines with inactivated TP53 in the validation set of cell lines considered to be TP53 wild type by Jeay et al. COSMIC database contains identical TP53 mutation(s) to the ones listed in Table 1 for each of these 8 cell lines. Therefore TP53 mutations listed in Table 1 of my manuscript are in complete agreement with corresponding entries in COSMIC. I’ve added [Supplementary-material SD1-data] which contains COSMIC information on mutations in cell lines listed in the Table 1 and also includes COSMIC sample ID for each of the 8 cell lines. COLO-818 is one of the two cell lines from Table 1 which does not have sequencing data in COSMIC, this cell line mRNA was sequenced by (Klijn et al., 2015) and it was found to contain C135R TP53 mutation, this is exactly the same mutation as listed for COLO-818 in Table 1.

References:

Klijn, C., Durinck, S., Stawiski, E.W., Haverty, P.M., Jiang, Z., Liu, H., Degenhardt, J., Mayba, O., Gnad, F., Liu, J., et al. (2015). A comprehensive transcriptional portrait of human cancer cell lines. Nat. Biotechnol. 33, 306–312.